# WHY DOES HIERARCHY (SOMETIMES) WORK SO WELL IN REINFORCEMENT LEARNING?

## ABSTRACT

Hierarchical reinforcement learning has demonstrated significant success at solving difficult reinforcement learning (RL) tasks. Previous works have motivated the use of hierarchy by appealing to a number of intuitive benefits, including learning over temporally extended transitions, exploring over temporally extended periods, and training and exploring in a more semantically meaningful action space, among others. However, in fully observed, Markovian settings, it is not immediately clear why hierarchical RL should provide benefits over standard "shallow" RL architectures. In this work, we isolate and evaluate the claimed benefits of hierarchical RL on a suite of tasks encompassing locomotion, navigation, and manipulation. Surprisingly, we find that most of the observed benefits of hierarchy can be attributed to improved exploration, as opposed to easier policy learning or imposed hierarchical structures. Given this insight, we present exploration techniques inspired by hierarchy that achieve performance competitive with hierarchical RL while at the same time being much simpler to use and implement.

## 1 INTRODUCTION

Many real-world tasks may be decomposed into natural hierarchical structures. To navigate a large building, one first needs to learn how to walk and turn before combining these behaviors to achieve robust navigation; to wash dishes, one first needs to learn basic object grasping and handling before composing a sequence of these primitives to successfully clean a collection of plates. Accordingly, hierarchy is an important topic in the context of reinforcement learning (RL), in which an agent learns to solve tasks from trial-and-error experience, and the use of hierarchical reinforcement learning (HRL) has long held the promise to elevate the capabilities of RL agents to more complex tasks (Dayan & Hinton, 1993; Schmidhuber, 1993; Parr & Russell, 1998; Barto & Mahadevan, 2003).

Recent work has made much progress towards delivering on this promise (Levy et al., 2017; Frans et al., 2018; Vezhnevets et al., 2017; Nachum et al., 2019). For example, Nachum et al. (2018a;b; 2019) use HRL to solve both simulated and real-world quadrupedal manipulation tasks, whereas state-of-the-art non-hierarchical methods are shown to make negligible progress on the same tasks. Levy et al. (2017) demonstrate similar results on complex navigation tasks, showing that HRL can find good policies with 3-5x fewer environment interactions than non-hierarchical methods.

While the empirical success of HRL is clear, the underlying reasons for this success are more difficult to explain. Prior works have motivated the use of HRL with a number of intuitive arguments: high-level actions are proposed at a lower temporal frequency than the atomic actions of the environment, effectively shortening the length of episodes; high-level actions often correspond to more semantically meaningful behaviors than the atomic actions of the environment, so both exploration and learning in this high-level action space is easier; and so on. These claims are easy to understand intuitively, and some may even be theoretically motivated (e.g., shorter episodes are indeed easier to learn; see Strehl et al. (2009); Azar et al. (2017)). On the other hand, the gap between any theoretical setting and the empirical settings in which these hierarchical systems excel is wide. Furthermore, in Markovian systems, there is no theoretical representational benefit to imposing temporally extended, hierarchical structures, since non-hierarchical policies that make a decision at every step can be optimal (Puterman, 2014). Nevertheless, the empirical advantages of hierarchy are self-evident in a number of recent works, which raises the question, why is hierarchy beneficial in these settings? Which of the claimed benefits of hierarchy contribute to its empirical successes?

In this work, we answer these questions via empirical analysis on a suite of tasks encompassing locomotion, navigation, and manipulation. We devise a series of experiments to isolate and evaluate the claimed benefits of HRL. Surprisingly, we find that most of the empirical benefit of hierarchy in our considered settings can be attributed to improved exploration. Given this observation, we propose a number of exploration methods that are inspired by hierarchy but are much simpler to use and implement. These proposed exploration methods enable non-hierarchical RL agents to achieve performance competitive with state-of-the-art HRL. Although our analysis is empirical and thus our conclusions are limited to the tasks we consider, we believe that our findings are important to the field of HRL. Our findings reveal that only a subset of the claimed benefits of hierarchy are achievable by current state-of-the-art methods, even on tasks that were previously believed to be approachable only by HRL methods. Thus, more work must be done to devise hierarchical systems that achieve *all* of the claimed benefits. We also hope that our findings can provide useful insights for future research on exploration in RL. Our findings show that exploration research can be informed by successful techniques in HRL to realize more temporally extended and semantically meaningful exploration strategies.

## 2 RELATED WORK

Due to its intuitive and biological appeal (Badre & Frank, 2011; Botvinick, 2012), the field of HRL has been an active research topic in the machine learning community for many years. A number of different architectures for HRL have been proposed in the literature (Dayan & Hinton, 1993; Kaelbling, 1993; Parr & Russell, 1998; Sutton et al., 1999; Dietterich, 2000; Florensa et al., 2017; Heess et al., 2017). We consider two paradigms specifically – the *options* framework (Precup, 2000) and *goal-conditioned* hierarchies (Nachum et al., 2018b), due to their impressive success in recent work (Frans et al., 2018; Levy et al., 2017; Nachum et al., 2018a; 2019), though an examination of other architectures is an important direction for future research.

One traditional approach to better understanding and justifying the use of an algorithm is through theoretical analysis. In tabular environments, there exist bounds on the sample complexity of learning a near-optimal policy dependent on the number of actions and effective episode horizon (Brunskill & Li, 2014). This bound can be used to motivate HRL when the high-level action space is smaller than the atomic action space (smaller number of actions) or the higher-level policy operates at a temporal abstraction greater than one (shorter effective horizon). Previous work has also analyzed HRL (specifically, the options framework) in the more general setting of continuous states (Mann & Mannor, 2014). However, these theoretical statements rely on having access to near-optimal options, which are typically not available in practice. Moreover, while simple synthetic tasks can be constructed to demonstrate these theoretical benefits, it is unclear if any of these benefits actually play a role in empirical successes demonstrated in more complex environments. In contrast, our empirical analysis is specifically devised to isolate and evaluate the observed practical benefits of HRL.

Our approach to isolating and evaluating the benefits of hierarchy via empirical analysis is partly inspired by previous empirical analysis on the benefits of options (Jong et al., 2008). Following a previous flurry of research, empirical demonstrations, and claimed intuitive benefits of options in the early 2000's, Jong et al. (2008) set out to systematically evaluate these techniques. Similar to our findings, exploration was identified as a key benefit, although realizing this benefit relied on the use of specially designed options and excessive prior knowledge of the task. Most of the remaining observed empirical benefits were found to be due to the use of *experience replay* (Lin, 1992), and the same performance could be achieved with experience replay alone on a non-hierarchical agent. Nowadays, experience replay is an ubiquitous component of RL algorithms. Moreover, the hierarchical paradigms of today are largely model-free and achieve more impressive practical results than the gridworld tasks evaluated by Jong et al. (2008). Therefore, we present our work as a re-calibration of the field's understanding with regards to current state-of-the-art hierarchical methods.

## 3 HIERARCHICAL REINFORCEMENT LEARNING

We briefly summarize the HRL methods and environments we evaluate on. We consider the typical two-layer hierarchical design, in which a higher-level policy solves a task by directing one or more lower-level policies. In the simplest case, the higher-level policy chooses a new high-level

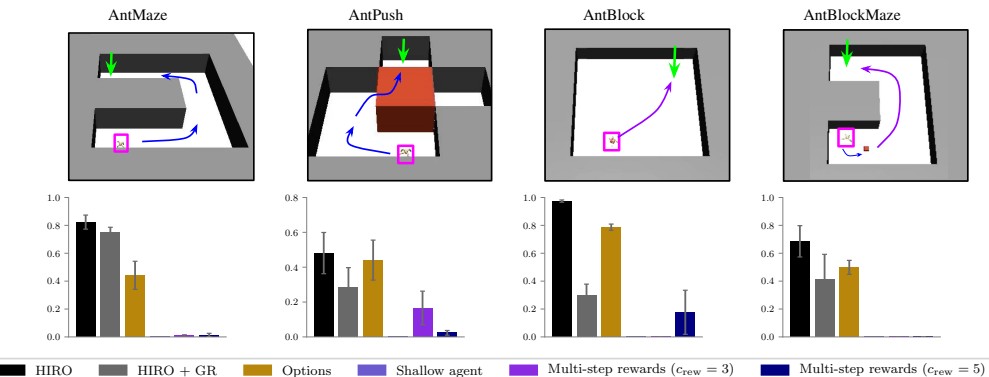

Figure 1: We consider four difficult tasks, where the agent (magenta) is a simulated quadrupedal robot. In AntMaze, the agent must navigate to the end of a U-shaped corridor (target given by green arrow); in AntPush, the agent must navigate to the target by first pushing a block obstacle to the right; in AntBlock and AntBlockMaze, the agent must push a small red block to the target location; see Nachum et al. (2018b) for more details. Task success rates are plotted for three HRL algorithms – HIRO (Nachum et al., 2018a), HIRO with goal relabelling (inspired by Levy et al. (2017)), and Options (Frans et al., 2018) – and shallow (non-hierarchical) agents with and without the use of multi-step rewards ($n$-step returns) over 10M training steps, averaged over 5 seeds. In this work, we isolate and evaluate the key properties of hierarchy which yield the stark difference in empirical performance between HRL and non-HRL methods.

action every $c$ timesteps.[1] In the options framework, the high-level action is a discrete choice, indicating which of $m$ lower-level policies (called options) to activate for the next $c$ steps. In goal-conditioned hierarchies, there is a single goal-conditioned lower-level policy, and the high-level action is a continuous-valued goal state which the lower-level is directed to reach.

Lower-level policy training operates differently in each of the HRL paradigms. For the options framework, we follow Bacon et al. (2017); Frans et al. (2018), training each lower-level policy to maximize environment reward. We train $m$ separate Q-value functions to minimize errors,

$$\mathcal{E}(s_t, a_t, R_t, s_{t+1}) = (Q_{\mathrm{lo},m}(s_t, a_t) - R_t - \gamma Q_{\mathrm{lo},m}(s_{t+1}, \pi_{\mathrm{lo},m}(s_{t+1})))^2, \qquad (1)$$

over single-step transitions, and the $m$ option policies are learned to maximize this Q-value $Q_{\mathrm{lo},m}(s_t, \pi_{\mathrm{lo},m}(s_t))$. In contrast, for HIRO (Nachum et al., 2018a) and HAC (Levy et al., 2017), the lower-level policy and Q-function are goal-conditioned. That is, a Q-function is learned to minimize errors,

$$\mathcal{E}(s_t, g_t, a_t, r_t, s_{t+1}, g_{t+1}) = (Q_{\mathrm{lo}}(s_t, g_t, a_t) - r_t - \gamma Q_{\mathrm{lo}}(s_{t+1}, g_{t+1}, \pi_{\mathrm{lo}}(s_{t+1}, g_{t+1})))^2, \qquad (2)$$

over single-step transitions, where $g_t$ is the current goal (high-level action updated every $c$ steps) and $r_t$ is an intrinsic reward measuring negative L2 distance to the goal. The lower-level policy is then trained to maximize the Q-value $Q_{\mathrm{lo}}(s_t, g_t, \pi_{\mathrm{lo}}(s_t, g_t))$.

For higher-level training we follow Nachum et al. (2018a); Frans et al. (2018) and train based on temporally-extended $c$-step transitions $(s_t, g_t, R_{t:t+c-1}, s_{t+c})$, where $g_t$ is a high-level action (discrete identifier for options, goal for goal-conditioned hierarchies) and $R_{t:t+c-1} = \sum_{k=0}^{c-1} R_{t+k}$ is the $c$-step sum of environment rewards. That is, a Q-value function is learned to minimize errors,

$$\mathcal{E}(s_t, g_t, R_{t:t+c-1}, s_{t+c}) = (Q_{\mathrm{hi}}(s_t, g_t) - R_{t:t+c-1} - \gamma Q_{\mathrm{hi}}(s_{t+c}, \pi_{\mathrm{hi}}(s_{t+c})))^2. \qquad (3)$$

In the options framework where high-level actions are discrete, the higher-level policy is simply $\pi_{\mathrm{hi}}(s) := \arg\max_g Q_{\mathrm{hi}}(s, g)$. In goal-conditioned HRL where high-level actions are continuous, the higher-level policy is learned to maximize the Q-value $Q_{\mathrm{hi}}(s, \pi_{\mathrm{hi}}(s))$.

Note that higher-level training in HRL is distinct from the use of *multi-step rewards* or *n-step returns* (Hessel et al., 2018), which proposes to train a non-hierarchical agent with respect to transi-

---

[1]We restrict our analysis to hierarchies using fixed $c$, although evaluating variable-length temporal abstractions are an important avenue for future work.

tions $(s_t, a_t, R_{t:t+c_{\text{rew}}-1}, s_{t+c_{\text{rew}}})$; i.e., the Q-value of a non-HRL policy is learned to minimize,

$$\mathcal{E}(s_t, a_t, R_{t:t+c-1}, s_{t+c}) = (Q(s_t, a_t) - R_{t:t+c_{\text{rew}}-1} - \gamma Q(s_{t+c_{\text{rew}}}, \pi(s_{t+c_{\text{rew}}})))^2, \qquad (4)$$

while the policy is learned to choose atomic actions to maximize $Q(s, \pi(s))$. In contrast, in HRL both the rewards *and* the actions $g_t$ used in the Q-value regression loss are temporally extended. However, as we will see in Section 5.2, the use of multi-step rewards alone can achieve almost all of the benefits associated with hierarchical training (controlling for exploration benefits).

For our empirical analysis, we consider four difficult tasks involving simulated robot locomotion, navigation, and object manipulation (see Figure 1). To alleviate issues of goal representation learning in goal-conditioned HRL, we fix the goals to be relative $x, y$ coordinates of the agent, which are a naturally good representation for our considered tasks. We note that this is only done to better control our empirical analysis, and that goal-conditioned HRL can achieve good performance on our considered tasks without this prior knowledge (Nachum et al., 2018b). We present the results of two goal-conditioned HRL methods: HIRO (Nachum et al., 2018a) and HIRO with goal relabelling (inspired by HAC; Levy et al. (2017)) and an options implementation based on Frans et al. (2018) in Figure 1. HRL methods can achieve strong performance on these tasks, while non-hierarchical methods struggle to make any progress at all. In this work, we strive to isolate and evaluate the key properties of HRL which lead to this stark difference.

## 4 HYPOTHESES OF THE BENEFITS OF HIERARCHY

We begin by listing out the claimed benefits of hierarchical learning. These hypotheses can be organized into several overlapping categories. The first set of hypotheses (H1 and H2 below) rely on the fact that HRL uses *temporally extended* actions; i.e., the high-level policy operates at a lower temporal frequency than the atomic actions of the environment. The second set (H3 and H4 below) rely on the fact that HRL uses *semantically meaningful* actions – high-level actions often correspond to more semantic behaviors than the natural low-level atomic actions exposed by the MDP. For example, in robotic navigation, the atomic actions may correspond to torques applied at the robot's joints, while the high-level actions in goal-conditioned HRL correspond to locations to which the robot might navigate. In options, there are many paradigms which are explicitly designed to achieve better exploration (McGovern & Barto, 2001; Kulkarni et al., 2016; Machado et al., 2017a;b). In the more undirected form of options that we use, it is argued that semantic behaviors naturally arise from unsupervised specialization of behaviors (Frans et al., 2018). The four hypotheses may also be categorized as *hierarchical training* (H1 and H3) and *hierarchical exploration* (H2 and H4).

**(H1) Temporally extended training**. High-level actions correspond to multiple environment steps. To the high-level agent, episodes are effectively shorter. Thus, rewards are propagated faster and learning should improve.

**(H2) Temporally extended exploration**. Since high-level actions correspond to multiple environment steps, exploration in the high-level is mapped to environment exploration which is temporally correlated across steps. This way, an HRL agent explores the environment more efficiently. As a motivating example, the distribution associated with a random (Gaussian) walk is wider when the random noise is temporally correlated.

**(H3) Semantic training**. High-level actor and critic networks are trained with respect to semantically meaningful actions. These semantic actions are more correlated with future values, and thus easier to learn, compared to training with respect to the atomic actions of the environment. For example, in a robot navigation task it is easier to learn future values with respect to deltas in x-y coordinates rather than robot joint torques.

**(H4) Semantic exploration**. Exploration strategies (in the simplest case, random action noise) are applied to semantically meaningful actions, and are thus more meaningful than the same strategies would be if applied to the atomic actions of the environment. For example, in a robot navigation task it intuitively makes more sense to explore at the level of x-y coordinates rather than robot joint torques.

Due to space constraints, see the Appendix for an additional hypothesis based on *modularity*.

## 5 EXPERIMENTS

Our experiments are aimed at studying the hypotheses outlined in the previous section, analyzing which of the intuitive benefits of HRL are actually present in practice. We begin by evaluating

the performance of HRL when varying the length of temporal abstraction used for training and exploration (Section 5.1, H1 and H2), finding that although this has some impact on results, it is not enough to account for the stark difference between HRL and non-hierarchical methods observed in Figure 1. We then look at the training hypotheses more closely (Section 5.2, H1 and H3). We find that, controlling for exploration, hierarchical training is only useful so far as it utilizes multi-step rewards, and furthermore the use of multi-step rewards is possible with a non-hierarchical agent. Given this surprising finding, we focus on the exploration question itself (Section 5.3, H2 and H4). We propose two exploration strategies, inspired by HRL, which enable non-hierarchical agents to achieve performance competitive with HRL.

## 5.1 EVALUATING THE BENEFITS OF TEMPORAL ABSTRACTION (H1 AND H2)

We begin by evaluating the merits of Hypotheses H1 and H2, both of which appeal to the temporally extended nature of high-level actions. In our considered hierarchies, temporal abstraction is a hyperparameter. Each high-level action operates for $c$ environment time steps. Accordingly, the choice of $c$ impacts two main components of learning:

- During training, the higher-level policy is trained with respect to temporally extended transitions of the form $(s_t, g_t, R_{t:t+c-1}, s_{t+c})$ (see Section 3 for details).
- During experience collection, a high-level action is sampled and updated every $c$ steps.

The first of these implementation details corresponds to H1 (temporally extended training) while the second corresponds to H2 (temporally extended exploration), and we can vary these two parameters independently to study the two hypotheses separately. Accordingly, we take HIRO, the best performing HRL method from Figure 1, and implement it so that these two instances of temporal abstraction are decoupled into separate choices $c_{\text{train}}$ for training horizon and $c_{\text{expl}}$ for experience collection horizon. We evaluate performance across different choices of these two hyperparameters.

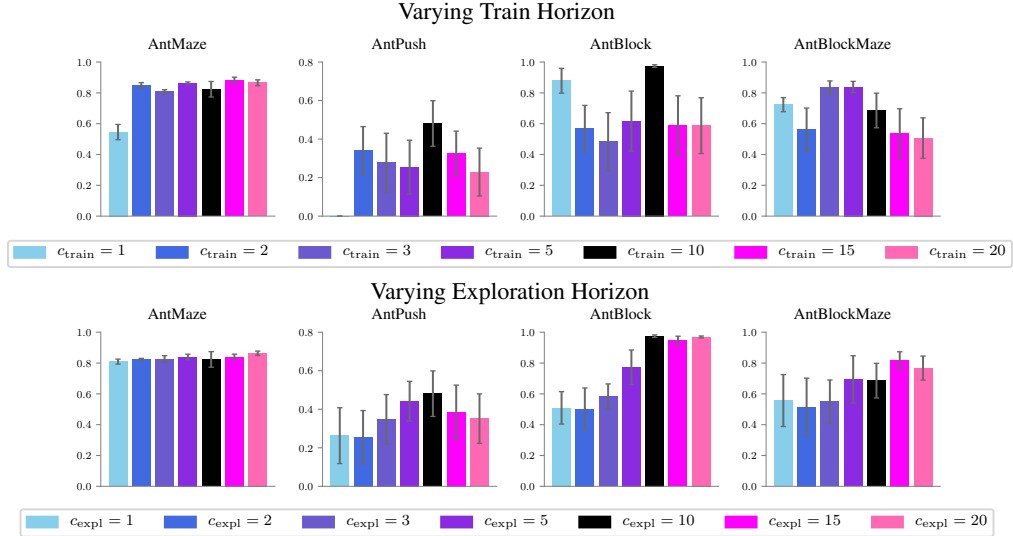

Figure 2: We present the results for different HRL methods while changing the temporal abstraction used for training ($c_{\text{train}}$, top) or the temporal abstraction used for experience collection ($c_{\text{expl}}$, bottom). Average success rates and standard errors are calculated for 5 randomly seeded runs, trained for 10M steps with early stopping. Recall that our HRL baselines use $c_{\text{train}} = c_{\text{expl}} = 10$. When varying $c_{\text{train}}$, we find that the choice of horizon matters only so far as $c_{\text{train}} > 1$. For $c_{\text{expl}}$, while there exists correlation between performance and temporal abstraction, using no temporal abstraction ($c_{\text{expl}} = 1$) can still make non-negligible progress compared to the shallow policies in Figure 1.

The results are presented in Figure 2, showing performance for different values of $c_{\text{train}}$ (top) and $c_{\text{expl}}$ (bottom); recall that our baseline HRL method uses $c_{\text{train}} = c_{\text{expl}} = 10$. The strongest effect of $c_{\text{train}}$ is observed in AntMaze and AntPush, where the difference between $c_{\text{train}} = 1$ and $c_{\text{train}} > 1$ is crucial to adequately solving these tasks. Otherwise, while there is some noticeable difference between specific choices of $c_{\text{train}}$ (as long as $c_{\text{train}} > 1$), there is no clear pattern suggesting that a larger value of $c_{\text{train}}$ is better.

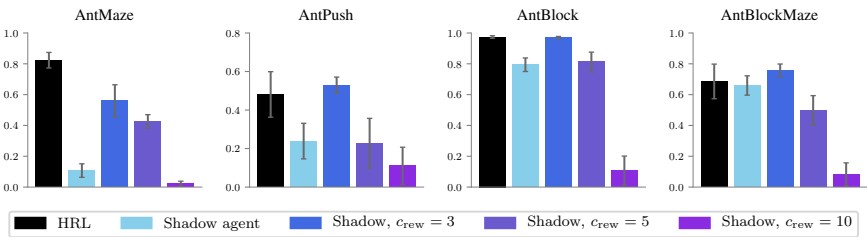

Figure 3: We evaluate and compare the performance of training a non-hierarchical *shadow* agent trained on experience collected by a hierarchical agent, thus disentangling the potential benefits of HRL for exploration from the potential benefits of HRL for training. In all environments except AntMaze, the shadow agent can achieve performance competitive with HRL, given an appropriate multi-step reward horizon ($c_{\mathrm{rew}} = 3$ performs best). Overall, this suggests that the effect of hierarchy on ease of training (as opposed to exploration) is modest, and can mostly be replicated by a non-hierarchical agent given good experience and the use of multi-step rewards.

For $c_{\mathrm{expl}}$, the effect seems slightly stronger. In AntMaze, there is no observed effect, while in AntPush, AntBlock, and AntBlockMaze there exists some correlation suggesting higher values of $c_{\mathrm{expl}}$ do yield better performance. Even so, $c_{\mathrm{expl}} = 1$ is often able to make non-negligible progress towards adequately solving the tasks, as compared to a non-hierarchical shallow policy (Figure 1).

Overall, these results provide intriguing insights into the impact of temporally abstracted training and exploration. While temporally extended training appears to help on these tasks, it is enough to have $c_{\mathrm{train}} > 1$. Temporally extended exploration appears to have a stronger effect, although it alone does not adequately explain the difference between an HRL agent that can solve the task and a non-hierarchical one that cannot make any progress. Where then does the benefit come from? In the next sections, we will delve deeper into the impact of hierarchy on training and exploration.

## 5.2 EVALUATING THE BENEFITS OF HIERARCHICAL TRAINING (H1 AND H3)

The previous section suggested that temporally extended training (H1) has at most a modest impact on the performance of HRL. In this section, we take a closer look at the benefits of hierarchy on training and study Hypothesis H3, which suggests that high-level actions used by HRL are easier for learning as compared to the atomic actions of the MDP. In goal-conditioned hierarchies for example, H3 claims that it is easier for RL to learn policy and value functions based on delta x-y commands (goals), than it is to learn policy and value functions based on atomic joint-torque actions exposed by the environment. In this section we aim to isolate this supposed benefit from other confounding factors, such as potential exploration benefits. Therefore, we devise an experiment to disentangle exploration from action representation, by training a standard non-hierarchical agent (a *shadow* agent) on experience collected by a hierarchical agent. If the benefits of HRL stem primarily from exploration, we would expect the shadow agent to do well; if representation of high-level actions matters for training, we would expected HRL to do better.

Accordingly, we augment our HRL implementation (specifically, HIRO) with an additional parallel shadow agent, represented by a standard single-level policy and value function. Each agent – the HRL agent and the non-hierarchical shadow agent – has its own replay buffer and collects its own experience from the environment. During training, we train the HRL agent as usual, while the shadow agent is trained on batches of experience gathered from *both* replay buffers (70% from the shadow agent's experience and 30% from the HRL agent's experience, chosen based on appropriate tuning). This way, any need for exploration is fulfilled by the experience gathered by the HRL agent. Will the non-hierarchical agent's policy still be able to learn? Or does training with a higher-level that uses semantically meaningful high-level actions make learning easier?

We present the results of our experiments in Figure 3. While the potential impacts of Hypotheses H2 and H4 (exploration) are neutralized by our setup, the impact of Hypothesis H1 (which Section 5.1 showed has a modest impact) still remains. As an attempt to control for this factor, we also consider a setup where the non-hierarchical shadow agent receives multi-step rewards (see Section 3 for an overview of multi-step rewards). Different temporal extents for the multi-step rewards are indicated by $c_{\mathrm{rew}}$ in the figure legend.

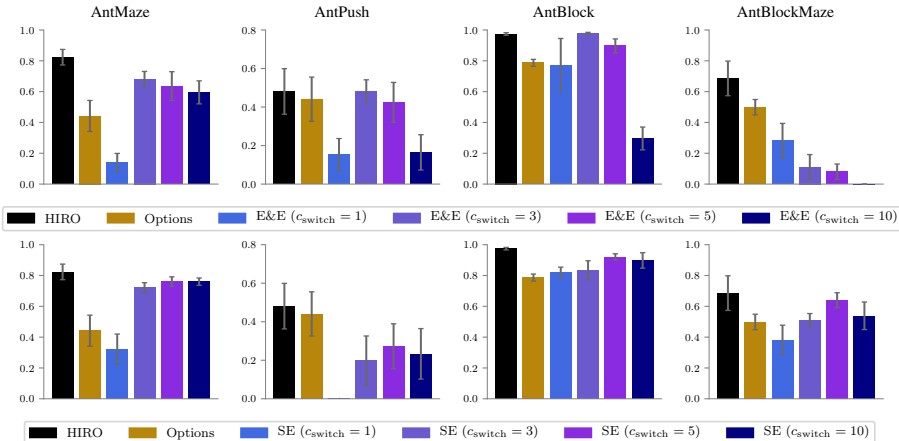

Figure 4: We compare the performance of HRL to Explore & Exploit (E&E) and Switching Ensemble (SE) – two non-hierarchical exploration methods that make use of HRL-inspired temporally extended modulation of behaviors (length of modulation given by $c_{\text{switch}}$). We find that the non-hierarchical methods are able to match the performance of HRL on these tasks (with the only exceptions being Explore & Exploit on AntBlockMaze and Switching Ensemble on AntPush), suggesting that exploration is the key to success on these tasks. These results also make clear the importance of temporally extended exploration; using $c_{\text{switch}} > 1$ is almost always better than $c_{\text{switch}} = 1$.

The results in Figure 3 show that learning from atomic actions, without higher-level action representations, *is* feasible, and can achieve similar performance as HRL. On AntMaze, we observe a slight drop in performance, but otherwise performance is competitive with HRL. The results across different multi-step reward horizons $c_{\text{rew}}$ also provide further insight into the conclusions of Section 5.1. As suggested by the results of Section 5.1, temporally abstracted training *does* affect performance, especially for AntMaze and AntPush. Still, while temporally abstracted training is important, these results show that the same benefit can be achieved by simply using multi-step rewards (which are much simpler to implement than using temporally extended actions). To confirm that multi-step rewards are not the only component necessary for success, see Figure 1, in which a non-hierarchical shallow agent with multi-step rewards is unable to make non-negligible progress on these tasks.

Overall, we conclude that the high-level action representations used by HRL methods in our domains are not a core factor for the success of these methods, outside of their potential benefits for exploration. The only observed benefit of high-level action representations in training is due to the use of multi-step rewards, and this can be easily incorporated into non-hierarchical agent training.

## 5.3 EVALUATING THE BENEFITS OF HIERARCHICAL EXPLORATION

The findings of the previous section show that training a non-hierarchical agent on 'good' experience (from a hierarchical agent) performs about as well as the hierarchical agent itself. If representing the policy and value function in terms of temporally extended, abstract actions is not crucial to achieving good performance, the next most-likely explanation is that the 'good' experience itself is the key. That is, good exploration is the key component to the success of HRL. This is the claim proposed by Hypotheses H2 (temporally extended exploration) and H4 (semantic exploration). In this section, we attempt to extend the experiments presented in Section 5.1 to better understand the impact of good exploration on the performance of a non-hierarchical agent. We will show that it is possible to enable non-hierarchical agents to achieve results competitive with HRL by using two exploration methods inspired by HRL: *Explore & Exploit* and *Switching Ensemble*.

Explore & Exploit is inspired by the hypothesis that goal-reaching is a good exploration strategy independent of hierarchy (Baranes & Oudeyer, 2010). Thus, we propose training two non-hierarchical agents – one trained to maximize environment rewards (similar to the higher-level policy in HRL), and the other trained to reach goals (similar to the lower-level policy in goal-conditioned HRL). Unlike in HRL, each policy operates on the atomic actions of the environments, and the goal for the

explore agent is sampled randomly according to an Ornstein-Uhlenbeck process[2] (standard deviation 5 and damping 0.8) as opposed to a learned policy. During experience collection, we randomly switch between the explore and exploit agents every $c_{\text{switch}}$ timesteps. Specifically, every $c_{\text{switch}}$ steps we randomly sample one of the two agents (with probability $0.2, 0.8$ for the explore and exploit agents, respectively), and this chosen agent is used for sampling the subsequent $c_{\text{switch}}$ atomic actions. Both agents share the same replay buffer for training. In this way, we preserve the benefits of goal-directed exploration – temporally extended and based on goal-reaching in a semantically meaningful space – without explicit hierarchies of policies.

Our other proposed exploration method, Switching Ensemble, is inspired by the options framework, in which multiple lower-level policies interact to solve a task based on their shared experience. We propose a simple variant of this approach that removes the higher-level policy. We train multiple (specifically, five) non-hierarchical agents to maximize environment rewards. During experience collection, we choose one of these agents uniformly at random every $c_{\text{switch}}$ timesteps. This way, we again maintain the spirit of exploration used in HRL – temporally extended and based on multiple interacting agents – while avoiding the use of explicit hierarchies of policies. This approach is related to the use of *randomized value functions* for exploration (Osband et al., 2014; 2016; Plappert et al., 2017; Fortunato et al., 2017) and may have a Bayesian interpretation (Gal & Ghahramani, 2016), although our proposal is unique for having a mechanism ($c_{\text{switch}}$) to control the temporally extended nature of the exploration. For both of these methods, we utilize multi-step environment rewards ($c_{\text{rew}} = 3$), which we found to work well in Section 5.2 (Figure 3).

Our findings are presented in Figure 4. We find that the proposed alternatives are able to achieve performance similar to HRL, with the only exceptions being Explore & Exploit on AntBlockMaze and Switching Ensemble on AntPush. Overall, these methods are able to bridge the gap in empirical performance between HRL and non-hierarchical methods from Figure 1, confirming the importance of good exploration on these tasks. Notably, these results show the benefit of temporally extended exploration even for non-hierarchical agents – using $c_{\text{switch}} > 1$ is often significantly better than using $c_{\text{switch}} = 1$ (switching the agent every step). Furthermore, the good performance of Explore & Exploit suggests that semantic exploration (goal-reaching) is beneficial, and likely plays an important role in the success of goal-conditioned HRL methods. The success of Switching Ensemble further shows that an explicit higher-level policy used to direct multiple agents is not necessary in these environments.

Overall, these results suggest that the success of HRL on these tasks is largely due to better exploration. That is, goal-conditioned and options-based hierarchies are better at exploring these environments as opposed to discovering high-level representations which make policy and value function training easier. Furthermore, these benefits can be achieved without explicit hierarchies of policies. Indeed, the results of Figure 4 show that non-hierarchical agents can achieve similar performance as state-of-the-art HRL, as long as they (1) use multi-step rewards in training and (2) use temporally-extended exploration (based on either goal-reaching or randomized value functions).

Beyond the core analysis in our experiments, we also studied the effects of modularity – using separate networks to represent higher and lower-level policies. Due to space constraints, these results are presented in the Appendix. These experiments confirm that the use of separate networks is beneficial for HRL. We further confirm that using separate networks for the Explore & Exploit and Switching Ensemble methods is crucial for their effectiveness.

## 6   DISCUSSION AND CONCLUSION

Looking back at the initial set of hypotheses from Section 4, we can draw a number of conclusions based on our empirical analysis. In terms of the benefits of training, it is clear that training with respect to semantically meaningful abstract actions (H3) has a negligible effect on the success of HRL (as seen from our *shadow* experiments; Figure 3). Moreover, temporally extended training (H1) is only important insofar as it enables the use of multi-step rewards, as opposed to training with respect to temporally extended actions (Figure 3). The main, and arguably most surprising, benefit of hierarchy is due to exploration. This is evidenced by the fact that temporally extended

---

[2]The use of OU noise for temporally-correlated exploration was first used by Lillicrap et al. (2015), where OU noise is added to the actions of a deterministic policy. In contrast, in our application, OU noise is used to determine a temporally correlated goal; i.e., the OU noise is used as input as opposed to added to the output of a policy.

| Hypothesis | Experiments | Important? |
|---|---|---|
| (H1) Temporal training | Figures 2, 3 | Yes, but only for the use of multi-step rewards ($n$-step returns). |
| (H2) Temporal exploration | Figures 2, 4 | Yes, and this is important even for non-hierarchical exploration. |
| (H3) Semantic training | Figure 3 | No. |
| (H4) Semantic exploration | Figure 4 | Yes, and this is important even for non-hierarchical exploration. |

Figure 5: A summary of our conclusions on the benefits of hierarchy.

goal-reaching and agent-switching can enable non-hierarchical agents to solve tasks that otherwise can only be solved, in our experiments, by hierarchical agents (Figure 4). These results suggest that the empirical effectiveness of hierarchical agents simply reflects the improved exploration that these agents can attain.

These conclusions suggest several future directions. First, our results show that current state-of-the-art HRL methods only achieve a subset of their claimed benefits. More research needs to be done to fully realize all of the benefits, especially with respect to semantic and temporally extended training. Second, our results suggest that hierarchy can be used as an inspiration for better exploration methods, and we encourage future work to investigate more variants of the non-hierarchical exploration strategies we proposed.

Still, our empirical analysis has limitations. Our results and conclusions are restricted to a limited set tasks and hierarchical designs. The use of other hierarchical designs may lead to different conclusions. Additionally, conclusions may be different for different task settings. For example, the use of hierarchy in multi-task settings may be beneficial for better *transfer*, a benefit that we did not evaluate. In addition, tasks with more complex environments and/or sparser rewards may benefit from other mechanisms for encouraging exploration (e.g., count-based exploration), which would be a complementary investigation to this study. An examination of different hierarchical structures and more varied settings is an important direction for future research.

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

## A    TRAINING DETAILS

We provide a more detailed visualization and description of HRL (Figure 6).

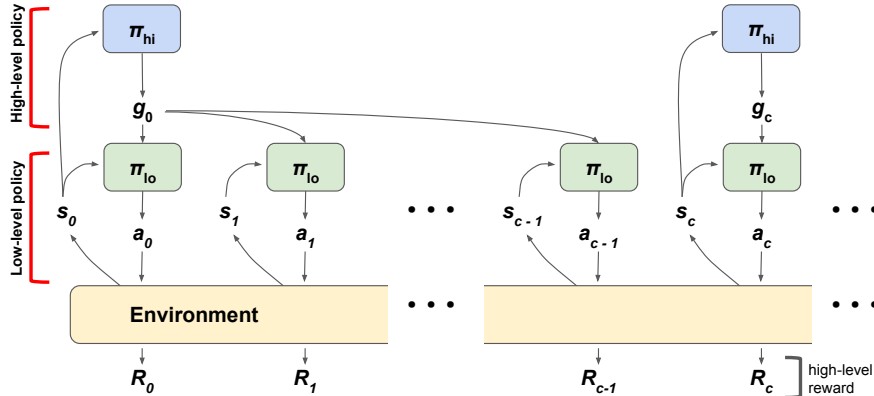

Figure 6: A diagram showing the structural form of an HRL agent. Every $c$ steps, a higher-level policy $\pi_{\text{hi}}$ chooses a high-level action $g_t$. In the options framework, this high-level action is an identifier, choosing which of $m$ options to activate. In goal-conditioned HRL, the high-level action is a goal-state. In either case, a lower-level policy $\pi_{\text{lo}}$ is used to produce a sequence of atomic actions $a_t$. After $c$ steps, control is returned to the higher-level policy.

## B    EVALUATING THE BENEFITS OF MODULARITY

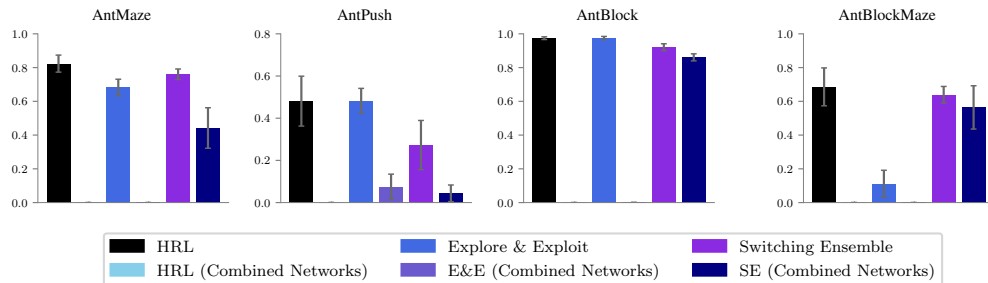

Figure 7: We evaluate the importance of modularity – in this case, using separate networks for separate policies. We find that using separate networks is consistently better, suggesting that modularity is important for both HRL and HRL-like methods.

We evaluate the merits of using *modularity* in HRL systems. We have already shown in the main text that a non-HRL agent can achieve performance similar to HRL. However, all of these non-HRL agents utilize multiple policies, similar to how HRL agents have separate lower-level and higher-level policies.

Thus, we evaluate how important this modularity is. We evaluate HIRO as well as the successful non-hierarchical methods from Section 5.3 with and without separate networks. Specifically, for HIRO we combine the separate networks for lower and higher-level policies into a single network with multiple heads. For Explore & Exploit and Five Exploit we combine the separate networks for each policy into a single network with multiple heads. The results are presented in Figure 7. We see that combined networks consistently lead to worse performance than structurally separate networks. HIRO and Explore & Exploit are especially sensitive to this change, suggesting that Hypothesis H5 is true for settings using goal-conditioned hierarchy or exploration. Overall, the use of separate networks for goal-reaching and task solving is beneficial to the performance of these methods in these settings.

## C    EXPERIMENT DETAILS

Our implementations are based on the open-source implementation of HIRO (Nachum et al., 2018a), using default hyperparameters. HIRO uses TD3 for policy training (Fujimoto et al., 2018), and so we train all non-hierarchical agents using TD3, with the same network and training hyperparameters as used by HIRO, unless otherwise stated.

Since the choice of $c$ in goal-conditioned HRL can also impact low-level training, as the frequency of new goals in recorded experience can affect the quality of the learned low-level behavior. To neutralize this factor in our ablations, we modify transitions $(s_t, g_t, a_t, r_t, s_{t+1}, g_{t+1})$ used for low-level training by replacing the next goal $g_{t+1}$ with the current goal $g_t$; in this way the lower-level policy is trained as if the high-level goal is never changed. This implementation modification has a negligible effect on HRL's performance with otherwise default settings.

To implement HIRO with goal relabelling, we augment the HIRO implementation with hindsight experience replay used for the lower-level policy. To implement Option-Critic (Bacon et al., 2017) in this framework, we create $m = 5$ separate lower-level policies trained to maximize reward (using $n$-step returns, where $n = 3$). We replace the higher-level continuous-action policy with a discrete double DQN-based agent, with $\epsilon$-greedy exploration ($\epsilon = 0.5$).

For our exploration alternatives (Explore & Exploit and Switching Ensemble), we utilize multi-step environment rewards with $c_{\text{rew}} = 3$, which we found to work well in Section 5.2 (see Figure 3). We also found it beneficial to train at a lower frequency: we collect 2 environment steps per each training (gradient descent) step. To keep the comparisons fair, we train these variants for 5M training steps (corresponding to 10M environment steps, equal to that used by HRL).

## D    ADDITIONAL EXPERIMENTS

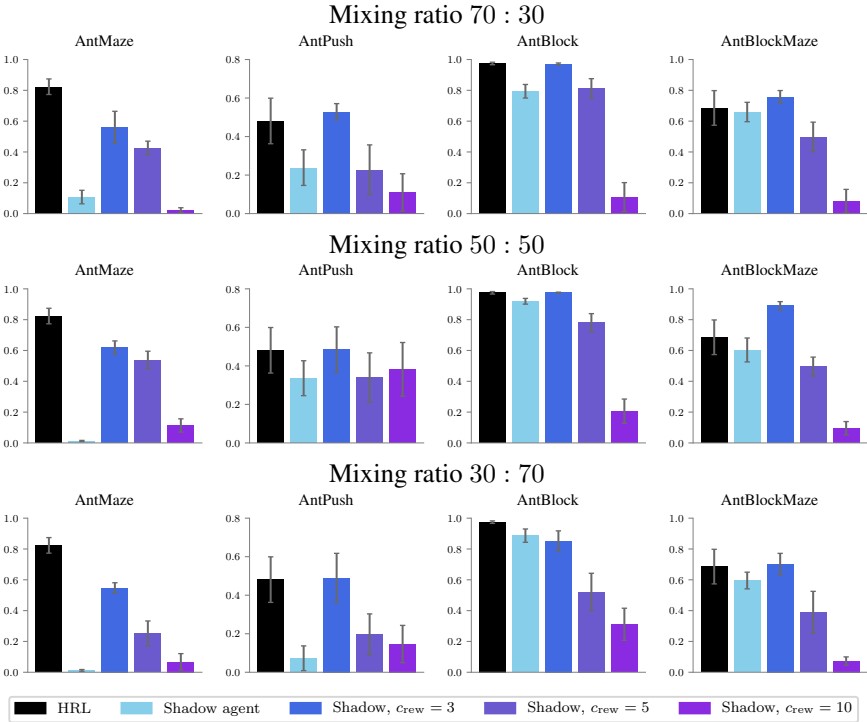

Figure 8:   We expand on the results from Figure 3, evaluating and comparing the performance of training a non-hierarchical *shadow* agent trained on experience collected by a hierarchical agent. Each row shows the results for a specific mixing ratio of experience between the shadow and HRL agents. We see that regardless of the mixing ratio, the conclusions are mostly consistent.

| Comparison | AntMaze | AntPush | AntBlock | AntBlockMaze |
|---|---|---|---|---|
| Figure 2, $c_{\text{train}} = 1$ vs. $c_{\text{train}} = 10$ | $(-)0.0054$ | $(-)0.0093$ | $(-)0.35$ | $(+)0.78$ |
| Figure 2, $c_{\text{expl}} = 1$ vs. $c_{\text{expl}} = 10$ | $(-)0.78$ | $(-)0.98$ | $(-)0.017$ | $(-)0.025$ |
| Figure 3, HRL vs. shadow with $c_{\text{rew}} = 3$ | $(+)0.088$ | $(-)0.73$ | $(+)0.73$ | $(-)0.60$ |
| Figure 3, shadow with $c_{\text{rew}} = 1$ vs. $c_{\text{rew}} = 3$ | $(-)0.014$ | $(-)0.046$ | $(-)0.022$ | $(-)0.29$ |
| Figure 4, HRL vs. E&E with $c_{\text{switch}} = 3$ | $(-)0.099$ | $(-)0.99$ | $(-)0.98$ | $(+)0.50$ |
| Figure 4, HRL vs. SE with $c_{\text{switch}} = 3$ | $(+)0.15$ | $(+)0.17$ | $(+)0.11$ | $(+)0.21$ |
| Figure 4, E&E with $c_{\text{switch}} = 1$ vs. $c_{\text{switch}} = 3$ | $(-)0.0002$ | $(-)0.023$ | $(-)0.35$ | $(+)0.29$ |
| Figure 4, SE with $c_{\text{switch}} = 1$ vs. $c_{\text{switch}} = 3$ | $(-)0.018$ | $(-)0.014$ | $(-)0.83$ | $(-)0.33$ |

Figure 9: Results of running a two-sided t-test on our experimental results. The sign designates the direction of the t-test result; i.e., a negative sign for A vs. B indicates that the mean of the results for A is less than that of B.

