# OpenReview forum: "Why Does Hierarchy (Sometimes) Work So Well in Reinforcement Learning?"
_ICLR.cc/2020/Conference — Reject_

### Official Review · AnonReviewer3 · 2019-10-18
**Official Blind Review #3**

**Rating:** 3

**Review:**

After Rebuttal:
After the discussion with the other reviewers, I tend to agree with them  regarding the need for more experimental evidence given the strong title and claims of the paper. For this reason I lower my  score from weak accept to weak reject. Please note that the idea that the paper tries to explore is of great value to the community and I encourage the authors to further perform more experiments (more environments and more algorithms) to solidify their claims.

************************************
The paper aims to answer the question proposed in the title. The authors conduct a series of experiments using well known hierarchical and non-hierarchical algorithms in order to extract the what is it that makes hierarchical reinforcement learning (HRL) efficient. The conclusion is that the main benefits of HRL are due to 1) temporally extended training enabling multi-step rewards, 2) temporally extended and semantically meaningful exploration.

I really enjoyed reading the paper and I believe it is an important contribution to the community. I believe the experiments are sufficient to support the hypothesis stated in the paper. Also, the authors state clearly the potential weak points of the paper in the discussion.  For the previous reasons I suggest the acceptance of this paper, although some improvements could be made.

Supporting arguments:

The paper is clearly well written. There is a clear exposition of the ideas, hypothesis and experiments.

The sequence of experiments makes sense to me. For example, section 5.1 tries to disentangle H1 and H2 (concluding both are important to different degrees), then in 5.2 H1 and H3 are evaluated (concluding that H3 is non-beneficial) and 5.3 focuses on the most important effect (exploration) through H2 and H4.

As mentioned on the discussion, other hierarchical designs, environments and tasks might lead to different conclusions. Even though this is true, I think the paper correctly balanced breadth vs depth of experiments and conditions. I think this paper serves as a baseline and might inspire further similar research that may cover wider settings (more hierarchical designs, environments, tasks etc.) to see if the results still hold.

Things to improve:


The data from all the figures presented in the paper could  be used to perform statistical tests to support the authors’ statements. For example, In Figure 2 (top) there is a varying parameter (c_train) and the authors claim that there is a noticeable effect on performance for c_train > 1. One can clearly see this in AntMaze and AntPush but it would be nice that when authors make such statements that they are backed by a significance test (e.g. analysis of variance, two-way anovas, t-tests, or any kind of test that the authors might deem suitable).  Similarly, one can do this for Figure 2(bottom),  and the remaining figures with their corresponding statements (e.g. c_switch >1 vs c_switch =1 in Figure 4).

Performing such statistical tests might require to run more than 5 seeds tough, but I believe the paper would be stronger with such statistical tests and I would increase the score accordignly.


**Experience Assessment:**

I have published one or two papers in this area.

**Review Assessment: Checking Correctness Of Derivations And Theory:**

N/A

**Review Assessment: Checking Correctness Of Experiments:**

I assessed the sensibility of the experiments.

**Review Assessment: Thoroughness In Paper Reading:**

I read the paper thoroughly.

---

> ### Author Response · Authors · 2019-11-12
> **Response**
>
> We thank the reviewer for their helpful feedback! Adding appropriate significance tests if a great idea. We have incorporated this into the paper (see Appendix D). Let us know if you have additional suggestions.

---

### Official Review · AnonReviewer1 · 2019-10-19
**Official Blind Review #1**

**Rating:** 3

**Review:**

This is an interesting paper, as it tries to understand the role of hierarchical methods (such as Options, higher level controllers etc) in RL. The core contribution of the paper is understand and evaluate the claimed benefits often proposed by hierarchical methods, and finds that the core benefit in fact comes from exploration. The paper studies hierarchical methods to eventually draw the conclusion that HRL in fact leads to better exploration based behaviour in complex tasks.

While the conclusion suggesting HRL leading to better exploration seems interesting, I am not sure whether this is in fact too surprising? For example, the Options framework has been fundamentally proposed and shows benefits in terms of transfer learning and faster exploration. Options has been fundamentally argued to lead to faster exploration (for example, when the goal changes in the four rooms task). Therefore, isn't the conclusion that this paper draws already known? Maybe the paper studies HRL methods in different experimental settings, not considerered in existing HRL or Options based frameworks before, but I would imagine similar conclusions can be drawn if a pure HRL system is studied?

The paper mainly does an experimental ablation study for HRL systems, and draws the conclusion of faster exploration led by HRL methods. However, I am not sure whether this conclusion, while assummably true as argued by previous methods too, is convincing enough based on the proposed suite of tasks?

It might have been more useful if there were similar theoretical contributions or analysis were also made for a stronger claim for HRL methods. Without lack of theoretical analysis or proofs, and simply based on the ablation studies as in this paper, I am not sure whether this paper is yet ready for acceptance in a venue like ICLR.

In the tasks considered, also there is no termination condition being learnt, and in fact the HRL systems studied here terminates every c time steps (as in HIRO and most of other HRL papers). This might be a significant limitation though. If we learn the termination condition, as in Option-Critic, would we expect similar behaviour in performance and can attribute the benefits of HRL only to exploration?

There are a vast amount of approaches based on identifying bottleneck states, and often they have shown better performance in terms of transfer learning. While in these approaches, often the benefits are claimed to be in terms of both exploration and transfer learning, this paper seems to contradict that?

Experimentally, there are only few specific tasks considered, like the variations of Ant tasks like AntPush and AntMaze. I am not convinced from the set of experimental results that they are sufficient enough to draw the conclusion that HRL methods only excel due to better exploration.

Why is the conventional Four Rooms domain ignored? What if we take the Four Rooms domain, change the goal states - I would expect the paper to do such analysis on the range of HRL methods, and then propose a convincing argument.

I think overall the paper needs more work, before such conclusion can be drawn overall. It seems to me like a strong claim that the benefits of HRL is only due to exploration. The paper does not do enough experimental abltation studies to strengthen the claim, mainly lacking fundamental HRL task setups. It does not do a theoretical analysis studying HRL methods and their benefits either. Overall, I think with these contributions, if similar behaviour persists, then it would make a more convincing argument.

There are a lot of theoretical papers studying provably sufficient exploration methods. Perhaps such approaches can also be taken here to study HRL methods and whether they provably lead to faster exploration?

Experimentally, I would expect a more wide range of task setups and domains to be studied - since this is mainly a paper based on experimental studies and trying to draw conclusion for existing HRL methods without proposing new approaches. I think without these carefully studied experiments, the conclusions are over-claimed.



**Experience Assessment:**

I have published one or two papers in this area.

**Review Assessment: Checking Correctness Of Derivations And Theory:**

I assessed the sensibility of the derivations and theory.

**Review Assessment: Checking Correctness Of Experiments:**

I carefully checked the experiments.

**Review Assessment: Thoroughness In Paper Reading:**

I read the paper thoroughly.

---

> ### Author Response · Authors · 2019-11-12
> **Response**
>
> We thank the reviewer for their thoughtful feedback. Our responses are below.
>
> “Therefore, isn't the conclusion that this paper draws [about exploration] already known?”
>
> — The hypotheses we evaluate are not new, but rather are based on often-claimed benefits of hierarchy.  Indeed, exploration is often cited as an advantage of hierarchy.  Still, the benefit of exploration is often claimed amongst other benefits regarding learning over abstract actions; e.g., the option-critic paper claims “Temporal abstraction is key to scaling up learning and planning in reinforcement learning”. A key contribution of our paper is empirically showing that exploration is the *key* benefit. In contrast, we show that learning over temporally or semantically abstracted actions has a modest effect, despite being often cited as an important benefit of HRL.  A complementary contribution of our paper is the exploration methods themselves — we show that simple exploration techniques inspired by hierarchy can solve difficult continuous-control problems. We hope that future work on exploration takes inspiration from these same ideas to develop better exploration methods in general.
>
> “lack of theoretical analysis or proofs, and simply based on the ablation studies”
>
> — The use of theoretical analysis as a way of motivating HRL is briefly covered in Section 2 of our paper. As stated there, “while simple synthetic tasks can be constructed to demonstrate theoretical benefits, it is unclear if any of these benefits actually play a role in empirical successes demonstrated in more complex environments.” In other words, theoretical statements can help motivate a method (and there are several previous works trying to do this for HRL), but it is hard to understand how much effect they have in practice, especially on the complex tasks we consider. For this reason, we focus our work on carefully devised empirical analyses.
>
> “There are only few specific tasks considered… Why is the conventional Four Rooms domain ignored?”
>
> — We chose our tasks based on the fact that they are previously considered to be only solvable by HRL methods. Prior to our work, non-HRL methods have been unable to make non-negligible progress on these tasks. In contrast, the “four rooms” domain is a simple task that is well-known to be solvable by many methods given appropriate tuning, and we believe this can lead a researcher to premature conclusions if tuning is not appropriately done. Jong 2008 “The Utility of Temporal Abstraction in Reinforcement Learning” has a few analyses on the four rooms domain as well as examples of incorrect conclusions by previous work based on poor experimental design in this setting (e.g., using or not using experience replay).

---

> > ### Comment · AnonReviewer1 · 2019-11-13
> > **Further Comments**
> >
> > I still think analysis on the four rooms domain can make the paper complete, and conclusions easier to draw for both the simple and complex task settings. While four rooms domain would need some fine tuning - this is true for any deep RL methods in general these days, so I don't think it should be ignored to avoid premature conclusions.
> >
> > The current analysis of the paper seems very focussed to certain types of HRL methods (as also pointed out by Reviewer 1), and I am not convinced that the conclusions this paper draws is true for any HRL methods in general.
> >
> > It would also be useful to add some theoretical analysis, justifying the results drawn in this paper? As you point out that several other works have done this previously, claiming that HRL methods can speed up learning (due to faster exploration) - it would perhaps be better to include some theoretical analysis that justifies/unjustifies the claims of this paper?
> >
> > My overall concern is : since the paper does experimental ablation studies on certain types of HRL algorithms, without theoretical justifications - I am reluctant to fully agree with the conclusions drawn in this paper. To me, it seems like there are some over claims made in the paper which should be toned down, otherwise it misleads the RL community about the overall benefits of HRL methods.

---

### Official Review · AnonReviewer2 · 2019-10-23
**Official Blind Review #2**

**Rating:** 3

**Review:**

This paper evaluates the benefits of using hierarchical RL (HRL) methods compared to regular shallow RL methods for fully observed MDPs. The goal of the work is to isolate and evaluate the benefits of using HRL on different control tasks (AntMaze, AntPush, AntBlock, AntBlockMaze). They find that the major benefit of HRL comes in the form of better exploration, compared to the ease of learning policies. They claim that the use of multi-step rewards alone is sufficient to provide the benefits associated with HRL. They also provide two exploration methods that are not hierarchical in nature but achieve similar performance:  a) Explore and Exploit and b) Switching Ensemble.

I appreciate the effort to contribute to the field by understanding and highlighting why HRL works better than shallow RL for a few tasks when theoretically there is no such incentive. However, this work still has a few missing components that need to be addressed in order to fully support the claims. Given these clarifications in an author's response, I would be willing to increase the score.


1) The claim needs more support.
The authors only compare policy-search based methods, on a very specific suite of control tasks, and then generalize the results. Maybe on these sets of tasks, exploration is the main issue, and not the training. This is without taking transfer or sparse-reward scenario into account. If noisy/sparse reward is added to the scenario, then training will also get difficult, and then maybe HRL methods can help with them too? Maybe not? But without considering that case, it seems wrong to come to a conclusion.

2) The hypothesis is only valid for HIRO.
In Sec 5, the authors test their method on a specific form of HRL method, but from Introduction and Conclusion, it seems that they are generalizing this conclusion to all HRL methods. The authors need to be either explicit about that these results only hold for HIRO, or provide some evidence that supports this generalization to other HRL methods.

3) Design choices not clear.
The following are the few choices that were made, but no argument or discussion regarding them was provided:
- In Sec 5.2, for the training of the shadow agent, why the particular split ratio (7:3) was used. Wouldn’t change in this ratio, can also lead to a different result for that comparison (H1 and H3)?
- In Sec 5.1, the train and explore hypothesis have c_{train} and  c_{explore} hyperparameters respectively, that control the multi-step horizon. However, how the exact decoupling is done is not clear. Is it that exploration sub-policies have a different horizon (c_{explore}) compared to how they are being updated, in terms of how the targets are calculated (c_{train}). With the nature of the algorithms based on policy-search methods, I don’t understand how this division induces decoupling between them.


4) Missing supporting literature/ Novelty
For the Explore and Exploit method, the authors propose a new method by adding OU noise and randomly switch between explore and exploit phase with c_{switch} hyperparameters. The use of OU noise to have temporally correlated exploration for continuous control tasks is already known [1]. Instead of explicitly exploring or exploiting [2], they use a hyper-parameter for the same, but no ablation study or discussion about the effect of that parameter is included. The same holds true for the Switching Ensemble case.




Things to improve the paper that did not impact the score:
Eq 3, shouldn’t there be discounting for g_{t} targets?







References:

[1] Lillicrap, Timothy P., et al. "Continuous control with deep reinforcement learning." arXiv preprint arXiv:1509.02971 (2015).

[2] Kearns, Michael, and Satinder Singh. "Near-optimal reinforcement learning in polynomial time." Machine learning 49.2-3 (2002): 209-232.


**Experience Assessment:**

I have published one or two papers in this area.

**Review Assessment: Checking Correctness Of Derivations And Theory:**

N/A

**Review Assessment: Checking Correctness Of Experiments:**

I assessed the sensibility of the experiments.

**Review Assessment: Thoroughness In Paper Reading:**

I read the paper thoroughly.

---

> ### Author Response · Authors · 2019-11-12
> **Response**
>
> Thanks for the careful reading of the paper and helpful feedback! Our responses are provided below:
>
> “Maybe on these sets of tasks, exploration is the main issue, and not the training.”
>
> — Our paper is based on an empirical analysis, and thus our conclusions may be different in other settings. While we attempted from the start to scope our claims appropriately, we understand that this may have not been communicated sufficiently.  We have thus further revised our writing by editing Sections 1 and 6 to make the limitations of our analysis clearer.
>
> Despite these limitations, our work still provides significant contributions. We consider tasks that were previously believed to be unapproachable by non-HRL methods, and show that they may be solved using much simpler alternatives (multi-step reward, switching ensembles, etc.). In addition to the impact this may have for the field of HRL, it also provides new avenues for exploration research. We show that simple exploration methods inspired by hierarchy can solve difficult and temporally extended continuous-control problems.
>
> “The hypothesis is only valid for HIRO.”
>
> — We also evaluate an options method based on the architecture used in meta-learned shared hierarchies and a HAC-based baseline (see Figure 1, yellow bar). We also attempted to solve these tasks with option-critic, but found its performance to be worse than our MLSH-style options baseline. Furthermore, many of our conclusions from Section 5 are independent of the use of HIRO. For example, in 5.2, the conclusion that a non-HRL agent can solve our considered tasks as long as it has access to good experience is true regardless of the fact that we specifically used HIRO to provide the “good experience”. As another example, in Section 5.3, the “switching ensemble” (Figure 4) is inspired by the options method. The success of switching ensemble suggests that for the MLSH-style options baseline, the use of a high-level policy is unnecessary (for these tasks). Instead, a hard-coded random-choice policy is enough for proper learning. Therefore, the conclusion is again that much of the complexity introduced by explicit hierarchies of learned policies is not necessary for good performance.
>
> “why the particular split ratio (7:3) was used”
>
> — The choice of this ratio is mostly arbitrary. For completeness, we have updated the paper to include results with other ratios (5:5 and 3:7, see Appendix D). Our conclusions are not sensitive to the specific ratio.
>
> (3) “how the exact decoupling is done is not clear”
>
> — C_explore determines how often the high-level is queried for a new action (goal) during data collection. Regardless of the value of c_explore, the replay buffer of experience will contain a sequence of steps (s_t,g_t,a_t,R_t).
>
> C_train determines the length of transitions used for high-level training (and only high-level training). That is, for high-level training we sample transitions of the form (s_t,g_t,R_{t:t+c_train-1},s_{t+c_train}). Implementation-wise, the choice of c_train affects (1) critic learning, via use of multi-step rewards and (2) both actor and critic learning via the relabelling of g_t in HIRO training: HIRO uses high-level goal relabelling, which can be interpreted as relabelling g_t to a goal \tilde{g}_t that is more appropriate for the training transition, despite the fact that c_explore != c_train.
>
> Each of these parameters can be chosen independently. Our experiments regarding c_train show that the use of high-level goals (i.e., learning over abstract actions) is not crucial to good performance. Rather, the use of multi-step rewards for critic learning is the main source of empirical benefit.
>
> “The use of OU noise to have temporally correlated exploration for continuous control tasks is already known”
>
> — Our exploration method is substantially different than traditional uses of OU noise. Specifically, the OU noise in our method is used to generate temporally correlated *goals*. These goals are then passed into the policies as input. This is in contrast to previous work, which simply adds OU noise to the *output* of a policy (anecdotally, we found that doing this more traditional form of exploration does not enable a non-HRL policy to solve the tasks). We will make sure to make a note of this in the paper and add a citation referring to the fact that OU noise has been used previously.

---

### Decision · Program_Chairs · 2019-12-19

**Decision:**

Reject

**Comment:**

This paper seeks to analyse the important question around why hierarchical reinforcement learning can be beneficial. The findings show that improved exploration is at the core of this improved performance. Based on these findings, the paper also proposes some simple exploration techniques which are shown to be competitive with hierarchical RL approaches.

This is a really interesting paper that could serve to address an oft speculated about result of the relation between HRL and exploration. While the findings of the paper are intuitive, it was agreed by all reviewers that the claims are too general for the evidence presented. The paper should be extended with a wider range of experiments covering more domains and algorithms, and would also benefit from some theoretical results.

As it stands this paper should not be accepted.